# Analgesic and Gastrointestinal Effects of Methadone in Horses Undergoing Orchiectomy

**DOI:** 10.3390/ani15162358

**Published:** 2025-08-11

**Authors:** Natalya Maldonado Moreno, Júlia Alves Moreira, Luiza Araujo De Oliveira, Amaranta Sanches Gontijo, Maria Luiza Castilho Baldi, Raphael Rocha Wenceslau, Andressa Batista da Silveira Xavier, Juan Felipe Colmenares Guzmán, Suzane Lilian Beier

**Affiliations:** Department of Clinical and Veterinary Surgery, Veterinary School, Universidade Federal de Minas Gerais, UFMG, Belo Horizonte 31270-901, MG, Brazil; natalyamaldonadomoreno@gmail.com (N.M.M.); juliaa.moreira@yahoo.com.br (J.A.M.); contatoluizaaraujo27@gmail.com (L.A.D.O.); amarantasg@gmail.com (A.S.G.); mariacastilhoobh@gmail.com (M.L.C.B.); rwenceslau@hotmail.com (R.R.W.); asilveiravet@gmail.com (A.B.d.S.X.); felipecolmenares94@gmail.com (J.F.C.G.)

**Keywords:** methadone, hypomotility, facial expression, abdominal ultrasound, horse

## Abstract

Adequate pain management is crucial in veterinary practice and is closely tied to animal welfare. In equines, however, it is sometimes overlooked due to concerns that opioids may cause central nervous system stimulation and increase the risk of intestinal hypomotility. This study evaluated the analgesic effect and the impact on intestinal motility of methadone in horses undergoing orchiectomy in a quadrupedal position. Nineteen male mixed-breed horses were divided into two groups and medicated with acepromazine (0.05 mg/kg) and detomidine (10 µg/kg); methadone (0.05 mg/kg) was administered to one group, saline to the other (ADM and ADS, respectively). Physiological variables, intestinal motility, gastric distention, and facial pain assessment were performed one day before, before the surgical procedure, and at 1, 2, 4, 6, and 8 h after administration of the ADM or ADS. Intestinal motility was reduced at two hours, returning to baseline between six and eight hours in both groups. Gastric dilatation was more pronounced in the methadone group, but facial pain scores were significantly lower. The methadone protocol, as suggested, may be effective for restraining horses and improving postoperative analgesia versus the ADS group. Although intestinal motility decreased, methadone did not worsen this, with similar reductions in both groups.

## 1. Introduction

Adequate pain management is crucial in veterinary practice and is closely tied to animal welfare. In equines, however, it is sometimes overlooked [1], particularly in routine procedures such as orchiectomy, which can generate intraoperative nociception and lead to complications and postoperative pain [2]. When considering sedation and general anesthesia in horses, a multimodal approach is often the most effective. This involves combining α2-adrenergic agonists, phenothiazines, and opioids to achieve optimal analgesic and sedative outcomes [3]. Detomidine and acepromazine are commonly used as pre-anesthetic agents in horses [4], and the combination of an α2 agonist and an opioid creates a synergistic effect, reducing the consumption of anesthetic agents and minimizing the side effects of each drug [3]. However, opioid use in horses is limited due to its potential for sympathetic stimulation, CNS stimulation, and excessive head movement [4,5], as well as its risk of inducing intestinal hypomotility [6,7]. Methadone, a synthetic μ agonist with similar analgesic potency and pharmacokinetics to morphine, is becoming increasingly popular due to its sedative, analgesic, and behavioral effects in horses [8].

Intestinal motility reflects gastrointestinal function and can be influenced by various factors, including anesthesia. The use of α2 agonists and opioids can induce hypomotility, causing predisposition to colic due to ileus and cecal impaction [9]. This depressant effect is mediated by the activation of receptors in the myenteric plexus, which alters gastrointestinal muscular contraction [9,10]. Auscultation of borborygmi and abdominal ultrasound are commonly used to assess the intestinal tract, presenting the advantage of being non-invasive. It is well established that α2 agonists [11] and opioids [5] reduce intestinal motility. However, previous studies [3,4,6,12] found no synergistic effect between them, meaning that combining the two drug classes did not result in further motility reduction. Although these effects have been investigated in conscious horses, the primary focus was on the antinociceptive and/or sedative properties of the methadone, using doses between 0.1 and 0.2 mg kg^−1^. The minimum clinically effective dose remains uncertain, as does its true effect on intestinal motility; therefore, further investigation is warranted.

The objective of this study was to evaluate the reduction in intestinal motility and gastric distension, as well as postoperative pain, during the 8 h period following administration of sedation protocols with or without methadone. We hypothesized that a low dose of methadone would not have a synergistic effect on the hypomotility induced by acepromazine and detomidine, while providing effective postoperative analgesia.

## 2. Materials and Methods

### 2.1. Animals and Experimental Design

This study is a randomized controlled clinical trial conducted at the Veterinary Hospital of the Federal University of Minas Gerais School of Veterinary Medicine. All pre- and postoperative assessments were performed by the same evaluator, who was blinded to the treatment protocol (ADM or ADS). This study was conducted with approval from the Animal Ethics Committee of UFMG under protocol 2/2023.

Nineteen male, non-castrated, client-owned horses of various breeds, aged between 2 and 6 years and weighing between 280 and 413 kg, were enrolled after obtaining informed consent from the owners. The horses had no history of gastrointestinal disorders and were considered healthy according to the American Society of Anesthesiologists (ASA) classification, based on physical examination as well as hematological and biochemical evaluations. All animals originated from the metropolitan region of Belo Horizonte and were referred to the Veterinary Hospital of the Veterinary School of UFMG for elective orchiectomy. The animals were hospitalized one day prior to the surgical procedure and kept in hospital stalls with access to commercial hay and water ad libitum to allow them to acclimate to the experimental environment.

The evaluations were performed in seven moments: one day before (DB), before the surgical procedure (BS), and at 1, 2, 4, 6, and 8 h (T1h, T2h, T4h, T6h, and T8h) after administration of the sedation protocols (ADM or ADS). In Figure 1, the progress through the study phases can be briefly found.

### 2.2. Surgical Procedure and Sedation Protocols

The horses were fasted for 12 h, with water provided ad libitum. On the day of the procedure, the animals were placed in a containment chute, where their weight was measured using a tape measure positioned at the withers. Following this, trichotomy and skin antisepsis were performed around the right jugular vein for intravenous catheterization with a 14-gauge, 45 mm catheter (Medix, Cascavel, Brazil). After venous puncture, a Luer lock device with closed access (PRN adapter, BD Althis, Rio do Sul, Brazil) was used to secure the catheter. Prophylactic treatments were then administered, including antitetanic serum (Vencosat; Dechra, Londrina, Brazil), 5000 international units (IU) via intramuscular (IM) injection, and potassium penicillin (Agrosil; Salud animal vansil, Descalvado, Brazil), 30,000 IU kg^−1^ via IM injection. Trichotomy and antisepsis of the surgical area were also performed.

The animals were randomly assigned to sedation protocols by a licensed veterinarian who selected a sealed envelope containing one of two options. For the ADM group, tranquilization was achieved with acepromazine (Apromazin 1%, Syntec, São Paulo, Brazil), 0.05 mg kg^−1^ IV, followed by sedation with detomidine (Detomidin 1%, Syntec, São Paulo, Brazil), 10 µg kg^−1^ IV over 30 s, five minutes later. After an additional five minutes, methadone (Mytedom 10 mg mL, Cristália, Itapira, Brazil), 0.05 mg kg^−1^, was administered over one minute. For the ADS group, methadone was replaced with saline solution (NaCl 0.9%, JP Farma, Ribeirão Preto, Brazil), administered via IV over one minute. The volume of the methadone and saline solution was matched to ensure the evaluator remained blinded to the treatment protocols. Before the procedure began, local anesthesia was applied using an intratesticular block and incisional block with lidocaine without vasoconstrictor (XYlestesin 2%, Cristália, Itapira, Brazil), with 15 mL injected into each testicle.

The orchiectomy was performed using the open technique [13] by two experienced surgeons who were unaware of the sedated protocol used. At the end of the surgery, the animals were administered meloxicam (Maxicam 2%, Ourofino Saude Animal, Cravinhos, Brazil), 0.6 mg kg^−1^ IV. They were then monitored in the containment chute for approximately two hours until they had fully recovered from anesthesia. After recovery, the animals returned to their stalls and were offered 2 kg of hay, three hours post-surgery, with free access to water.

### 2.3. Physiological Parameters

Heart rate (HR) was measured by auscultation of the left thoracic region between the fourth and fifth intercostal spaces, at the level of the ulnar–radioulnar–humeral joint. Respiratory rate (fR) was assessed by observing the movement of the costal muscles. Rectal temperature (RT, °C) was recorded using a digital thermometer. Mucous membrane color (MUC) and capillary refill time (CRT) were evaluated by visual inspection and digital pressure of the mucous membranes.

### 2.4. Intestinal Motility Evaluation/Abdominal Auscultation

Intestinal motility was assessed by auscultating intestinal sounds with a stethoscope in four abdominal quadrants. The right dorsal quadrant (RDQA) was auscultated in the paralumbar fossa to evaluate the ileocecal valve, cecum, and right dorsal colon, while the right ventral quadrant (RVQA) was auscultated along the edge of the last three ribs to assess the right ventral colon. The same procedure was performed on the left side, with the left dorsal quadrant (LDQA) examined to evaluate the left dorsal colon and small intestine in the caudal portion of the paralumbar fossa and the left ventral quadrant (LVQA) examined to assess the left ventral colon in the ventral portion of the abdomen. Each quadrant was auscultated for one minute by a blinded evaluator, and each quadrant received an intestinal motility score based on validated guidelines [14] as follows: 0—no borborygmus; 1—the period without borborygmus was longer than the period with borborygmus; 2—the period with borborygmus was longer than the period without them; 3—constant and continuous borborygmus. A score of 2 was considered closest to physiological conditions.

### 2.5. Intestinal Motility Evaluation/Abdominal Ultrasound Evaluation

The ultrasound evaluation was performed by the same blinded observer who conducted the auscultation assessment, following established guidelines [7,15,16,17], using a macroconvex transcutaneous ultrasonograph transducer with a frequency of 2.5–5 MHz (Mindray^®^ M5, Bio-Medical Electronics Co. Ltd., Shenzhen, China). Alcohol was applied to ensure better contact between the skin and the transducer. Four intestinal segments were evaluated: descending duodenum (USDUO), body of the cecum (USCEC), right ventral colon (USRVC), and left ventral colon (USLVC) (Figure 2). When the image was deemed optimal, a two-minute scan was recorded for each segment. The number of contractions per minute for each segment was then calculated by the blinded observer.

The size of the stomach (EST) was also measured by scanning between the 8th and 13th intercostal spaces on the left side, identifying the most caudal space where the dorsal face of the large curvature was visible. The stomach was recognized by its wall thickness and smooth, hyperechogenic, and round appearance. The left ventral colon (LVC) was scanned ventrally in the paralumbar region on the left side of the abdomen, along the line extending from the knee ventral to the spleen. The right ventral colon (RVC) was scanned ventrally in the paralumbar region on the right side of the abdomen, following the same line. Both segments were identified by their hyperechoic, shiny walls with the presence of sacculation. Contractions of the LVC and RVC were evaluated based on changes in their shape. The descending duodenum was scanned in the right thoracic region between the 8th and 10th ribs, along the line connecting the olecranon to the coxal tuberosity. It was visualized as a flattened hyperechoic line that changed to an oval or round shape during peristaltic movement, positioned between the liver and the right dorsal colon, adjacent to the caudal pole of the right kidney. Lastly, the body of the cecum was identified in the dorsal paralumbar region on the right side. Contractions of the cecum were noted by changes in the movement of its wall, as visualized on the transducer, with a contraction amplitude greater than 2 cm.

### 2.6. Postoperatory Pain Evaluation

Postoperative pain was evaluated using the Utrecht University guideline for Facial Evaluation of Pain in Equines (EQUUS-FAP), as described by Van Loon and Van Dierendonck 2015 [18]. Each of the nine parameters was scored from 0 to 2, from a distance of approximately two meters to avoid interfering with the horses’ behavior. The assessment was conducted before auscultation and ultrasound examination by two experienced veterinarians who were blinded to treatment. The total pain score was calculated by summing the individual scores, where 0 indicated no signs of pain and 18 represented the worst possible pain. If a horse scored over seven points on the scale, it was administered analgesic rescue with methadone at 0.1 mg kg^−1^ IM.

### 2.7. Statistical Analysis

The results are presented as mean and standard deviation or median and interquartile range (Q1–Q3). For the variables HR, fR, and RT, analysis of variance was performed using an adjusted linear mixed model that accounted for the fixed effects of treatment, time, and their interaction, as well as the random effect of animal selection to address the repeated measures over time. The homogeneity of variance and residual normality were assessed through graphical analysis of the residuals. For variables that did not meet the assumptions for analysis of variance, the Mann–Whitney test was applied to evaluate the effect of treatments within each time point, and the Durbin test was used to assess the effect of time within each treatment. In order to understand power of hypothesis tests for the most important evaluated traits, post hoc power analysis was performed for the contractions per minute measured by abdominal ultrasound. A significance level of 5% was considered, and all analyses were conducted using R software 4.3.1 (R Core Team 2023; Bell Laboratories, Windsor, WI, USA) [19].

## 3. Results

The study sample consisted of 19 male, non-castrated horses from various breeds, with an average age of 4.2 ± 2 years and weight of 345.3 ± 67.7 kg. All the horses completed this study without complications and did not show any signs of acute abdominal pain or excitation.

For the physiological parameters evaluated—HR, fR, and RT—no significant differences were observed between the ADM and ADS groups. However, differences over time within each group were noted (Table 1). Both HR and fR remained within the species’ physiological range, but the ADM group showed less variability over time compared to the ADS group. This difference was most evident at 1 and 2 h after the surgical procedure. The RT of the animals remained normal for the species, except at T1h and T2h, when it dropped below 37.5 °C in both the ADM and ADS groups. Regarding capillary refill time, no significant differences were observed between the groups. The animals showed normochromic mucosae and capillary refill times (CRT) of 2 s up to 8 h after the procedure.

In this study, fasting (BS) did not appear to influence motility scores when compared to animals with ad libitum access to food (DB), and values recorded at both time points were consistent with the physiological parameters expected for the species. One hour after protocol administration (T1h), both groups exhibited a significant decrease in intestinal motility. In the ADM group, the motility scores returned to baseline at T2h and were slightly elevated at T4h, whereas the values at T6h and T8h remained within the physiological range. In the ADS group, motility remained decreased until T2h, before returning to physiological levels at T4h, T6h, and T8h (Table 2).

At LDQA, the motility score differed between the groups at T1h (*p* = 0.0065), with ADS displaying lower values (median 0–1) compared to ADM (median 1–2) and T4h (*p* = 0.0344) (2–2 versus 2–2.75), respectively. The same pattern was observed in the RDQA quadrant at T1h (*p* = 0.0302) and T4h (*p* = 0.0441). There were also differences in motility over time between the ADM (*p* <0.001) and ADS (*p* = 0.0002) treatments. At LVQA, there were significant differences between the groups at T2h (*p* = 0.0428), with ADS displaying lower values (median 1–2) compared to ADM (median 2–2) and T4h (*p* = 0.0060) (2–2 versus 2–3), respectively, as well as differences within the groups over time. Finally, the RVQA scores were significantly different between groups at T1h (*p* = 0.0191), with ADS displaying lower values (median 1–1) compared with ADM (median 1–2). All other intervals were considered to be within the physiological range (Table 2).

The USRVC showed a significant difference between the groups at T6h (*p* = 0.0788), with the ADS group exhibiting fewer contractions per minute compared to the ADM group. This difference was not observed in the equivalent anatomical region by auscultation of the RVQA. The USLVC did not show a significant difference between the groups (*p* > 0.05); however, there was a significant difference across the evaluated time points within the ADS group (*p* = 0.045). At T1h, a reduction in contractions per minute was observed compared to DB (median 1.0 contraction/min). Motility returned to normal values at the subsequent time points. The USCEC presented a significant difference between the groups at T6h (*p* = 0.0499), which was not found in the corresponding anatomical region by auscultation of the RDQA. At T1h, the ADM group showed fewer contractions per minute compared with the ADS group. The values at DB and T4h in the ADM group were higher, while at BS, T2h T6h, and T8h, the motility remained within the physiological parameters. Finally, the USDOU did not show a significant difference between the ADM and ADS treatments, and no significant differences were observed between the intervals within each group (Table 3).

There was a significant difference when comparing the two groups, ADM and ADS, regarding stomach size at DB, T4h, T6h, and T8h (*p* = 0.0079, *p* = 0.0160, *p* = 0.0446, and *p* = 0.0064). Specifically, in the ADM group, the stomach was observed at the 15th intercostal space at T8h, which was three more intercostal spaces compared to DB (considered physiological) and five more compared to BS (fasting). In the ADS group, displacement of the greater curvature of the stomach was also observed on ultrasound, although this change was more subtle. The stomach was visualized at the 11th intercostal space at T8h, one more space than at DB and two more than at BS (Table 4).

None of the patients in either the ADM or ADS groups required analgesic rescue during the 8 h of the experiment. However, a significant statistical difference was observed between the groups at T2h (*p* = 0.0138), T4h (*p* = 0.0002), and T6h (*p* = 0.0104). Similarly, when comparing the intervals within each treatment, significant differences were found for the ADM group (*p* < 0.001) and the ADS group (*p* < 0.001) at T2h, T4h, and T6h (Table 5). The median values in the ADM group remained between 0 and 2 throughout the evaluations, while in the ADS group, medians ranged from 1 to 4. This likely occurred because, during the evaluation, the animals in the ADM group appeared more comfortable, as indicated by their facial expressions, compared to those in the ADS group.

## 4. Discussion

In this study, no statistically significant differences were observed between the groups for FC and Fr (*p* > 0.05). However, the ADM group (acepromazine 0.05 mg kg^−1^, detomidine 10 µg kg^−1^, and methadone 0.05 mg kg^−1^) exhibited lower variability in HR and fR over time compared with the ADS group (acepromazine 0.05 mg kg^−1^, detomidine 10 µg kg^−1^, and saline solution NaCl 0.9%). It was also noted that HR and fR values were lower at 1 and 2 h after the procedure, which may be explained by the duration of action of the drugs. α2 agonists, when administered alone, are often associated with reduced HR, decreased cardiac output, and increased pulmonary and systemic vascular resistance [11]. Initially, this response is due to a reflex mediated by baroreceptors (vagal), which increases the systemic vascular resistance. As the response normalizes, persistent bradycardia is frequently linked to decreased central sympathetic activity [11]. This effect may also result from the action on presynaptic α2-adrenergic receptors, reducing the release of noradrenaline, inhibiting synaptic tone, and enhancing parasympathetic nervous system activity [20].

Previous studies have shown a reduction of 30% to 50% in HR with the use of detomidine at doses ranging from 10 to 20 µg kg^−1^ in healthy adult horses [21,22]. In this study, the changes were less than 20% one and two hours after the procedure, which is consistent with previous findings [23]. Additionally, the reduction in fR might be explained by the central action of α2 agonists [21], as suggested by a previous study [23], which reported a decrease in fR one hour after administering detomidine at a dose of 20 µg kg^−1^. Finally, regarding opioids, fR increases when used in conscious horses due to their excitatory effects on the central nervous system. These alterations are generally not significant and depend on the specific drug and its method of administration [5]. In this study, the use of methadone at a dose of 0.05 mg kg^−1^ IV did not result in a clinically relevant increase in fR, HR, or behavioral effects. This may be explained by the concurrent administration of acepromazine and detomidine, which depress dopaminergic and noradrenergic transmission, respectively [5]. Moreover, opioid alone does not counteract the central depressive effects of α_2_ agonists; therefore, reductions in HR and fR may occur, as observed during the first two hours. These findings are consistent with previously published studies using such combinations, which consistently demonstrate that the hemodynamic effects of α_2_ agonists prevail over those of opioid analgesics [5,6,8].

Prior studies [2,24] have shown that castration is associated with a certain level of pain, thus justifying the use of analgesics. In the present study, the inclusion of methadone in the neuroleptanalgesia protocol resulted in lower scores on the EQUUS-FAP scale compared to the group that did not receive the opioid. Various pain assessment scales have been developed for horses, and those based on facial expression in particular have demonstrated promising results in identifying pain in horses post-castration [25], visceral pain [26], acute pain, or postoperative pain on the head [27,28]. The EQUUS-FAP scale used in the present study allowed for the observation of significant differences between groups and across the different intervals evaluated, based on changes in facial expression. It has previously been demonstrated that pain in horses can be expressed through general, non-specific indicators, such as reduced normal activity, lowered head, fixed gaze, rigid posture, and reluctance to move [29]. Three of the nine evaluated parameters were more consistently observed in the present study (reduced head movement, less alert state, and ear movement towards sound), aligning with previous studies [25,29], which suggest a decrease in behaviors such as exploration, less alert state, and lowered head with minimal movement in patients showing pain.

The results of this experiment reinforce the enhanced analgesic effect provided by the combination of acepromazine, detomidine, and methadone (ADM). This effect is likely attributable not only to methadone’s intrinsic analgesic properties but also to its potential interactions with detomidine (positive potentializing and additive synergism) [4,8,30]. In a study where horses received acepromazine 0.05 mg kg^−1^ + detomidine 15 µg kg^−1^ in association with levomethadone 100 µg kg^−1^ and undergoing dentary extraction, postoperatively presented lower EQUUS-FAP scores when compared to animals who received buprenorphine 5 µg kg^−1^ or butorphanol 100 µg kg^−1^ [31]. Similar results were observed by other authors, when detomidine 5 µg kg^−1^ combined with methadone 0.2 mg kg^−1^ induced antinociceptives effects more intense and persistent, when compared to detomidine by itself in low or high doses of 2.5 and 5 ug kg^−1^ or Nacl 0,9% for three nociceptive stimuli (thermic, mechanic, and electric) [12]. In the same way, the association of detomidine 10 ug kg^−1^ with methadone 0.2 mg kg^−1^ induced more intense and prolonged antinociception, in contrast, when the dose of detomidine was reduced to 5 ug kg^−1^; the antinociception was similar but shorter for thermic, electric, and mechanic stimuli [6].

In the ADM group, a single preoperative IV dose of methadone provided analgesia for the entire 8 h duration of the experiment. This effect can be attributed to the action of methadone in equines when administered parenterally, with the onset of action being between 10 and 20 min and a duration of 4 to 8 h [22]. In this study, no increase in spontaneous locomotor activity, excitation, or head movement was observed in the ADM group, which is often seen after the use of opioids [3,8]. This likely occurred due to the combined use of acepromazine and detomidine [4], which are known to mitigate these symptoms. Furthermore, the low dose of methadone used (0.05 mg kg^−1^) may have contributed to this outcome, as higher doses (0.1, 0.5, and 1.0 mg kg^−1^) administered intravenously have been reported to increase locomotor activity and CNS stimulation. These effects typically begin within 5 min of administration, peak at 30 min, and last for 1 to 4 h [8]. In addition, animals in the ADM group appeared more sedated and relaxed during and after the procedure, which can be considered a beneficial clinical outcome, essential to ensuring patient welfare, reducing morbidity, and optimizing recovery.

Regarding gut motility in horses, auscultation [14,32] and ultrasonography [7,9,14,33] are methods commonly used to assess gastrointestinal motility. However, the data obtained through ultrasonography are quantitative [16], while auscultation provides subjective evaluations [32]. Despite this, successful analyses have been reported using subjective scores for abdominal sounds [10,14]. The present study showed a decrease in intestinal motility in both groups following premedication, with reductions observed between 60 and 120 min for the ADM and ADS groups, respectively. These findings were expected, as the protocols included opioids, tranquilizers, and α2 agonists, which are known to induce such an effect [9,10]. However, no clinical signs of abdominal discomfort (colic) were observed in any of the animals during the observation period. Hypomotility associated with opioids is linked to their action on opioid receptors in both the central nervous system and the gastrointestinal tract [10,34]. This can lead to constipation, adynamic ileus, increased tone in intestinal smooth muscle, and inhibition of the mechanical processes responsible for propulsive motility [10].

Specifically, detomidine stimulates the α2 receptors located in the myenteric plexus, inhibiting acetylcholine release and consequently reducing myenteric intestinal activity [35]. However, in a previous study, the combination of detomidine and methadone did not produce an additive effect, meaning there was no significant further reduction in intestinal motility when comparing isolated use to combination use of these drugs. In that study, intestinal motility was reduced for 150 min after a combination of detomidine and methadone (2.5 µg kg^−1^ + 0.2 mg kg^−1^) and 210 min after a higher dose of detomidine (5 µg kg^−1^) or the combination of detomidine and methadone (5 µg kg^−1^ + 0.2 mg kg^−1^) [12]. In another study by the same author, intestinal motility, assessed by auscultation, was reduced for 30 min after administration of methadone (0.2 mg kg^−1^), detomidine (5 µg kg^−1^), or the combination of methadone with detomidine (2.5 or 5 µg kg^−1^) and up to 75 min after the administration of methadone (0.2 mg kg^−1^) and detomidine (10 µg kg^−1^) [6]. The findings from these studies suggest that methadone has a minimal or additive effect on reducing motility, whereas detomidine induces a dose-dependent decrease in motility. In our results, motility remained similarly reduced, between 1 and 2 h in the methadone (ADM) group compared to the group without methadone (ADS), respectively. This indicates that the effects of methadone on motility were minimal at the dose used.

In the present study, the inclusion of methadone did not present a risk to the patients, as intestinal motility returned to physiological values, with the reduction being temporary. Previous studies [8,10] indicated that hypomotility is inversely proportional to the dose of opioids administered, with more pronounced and longer-lasting effects at doses of 0.5 mg kg^−1^ IV of methadone or morphine. In contrast, the combination of detomidine (10 µg kg^−1^) and methadone (0.2 mg kg^−1^) IV, as well as detomidine alone, resulted in a significant reduction in gastrointestinal motility compared to physiological values, between 30 min and 1 h after administration, with no significant difference between the two groups [6]. These findings are consistent with the results of our study, where, in both treatments, intestinal motility scores were reduced one hour after the administration of the protocols. However, motility recovery was slower in the ADS group compared to the ADM group, suggesting that the dose of 0.05 mg kg^−1^ of methadone had little influence on motility and that the isolated use of α2 agonists was the main factor responsible for the observed outcome due to their suppressive effect [6,12]. In addition, pain itself may have a more profound impact on gut function, potentially inducing hypomotility [36]. In this study, adequate pain control may have contributed to the recovery of intestinal motility in the ADM group compared with the ADS group, a finding supported by the facial expression scores.

No studies were found evaluating intestinal motility via ultrasound with the use of methadone in horses. Nevertheless, the gastrointestinal side effects of morphine use have already been studied in healthy horses through clinical and ultrasonographic evaluation following a 0.1 mg kg^−1^ dose [7]. In this study, the number of contractions in the duodenum, cecum, and right and left ventral colon, as well as the scores of intestinal borborygmus, were significantly reduced 1 h after administration. In another study [33], cecal activity was not significantly affected by fasting or xylazine sedation. However, in fasted horses, xylazine (0.4 mg kg^−1^) significantly decreased jejunal and cecal activity. In the present study, the USLVC and USCEC detected a difference between intervals in the ADS group (*p* = 0.045), with reduced intestinal motility observed at T1h, one hour after detomidine administration, in comparison with the physiological baseline (DB). A similar result was observed by auscultation in the corresponding area of the USLVC (LVQA). The USCEC showed a significant difference between the groups at T6h (*p* = 0.0499), where the ADM group had an increase in contractions per minute, while the ADS group exhibited a decrease. This was not observed by auscultation in the corresponding anatomical region (RDQA); nevertheless, this finding was not associated with any clinically significant consequences. The USRVC did not show a decrease in motility (contractions per minute) and did not align with the auscultation method for the corresponding anatomical region (RVQA), where the score was reduced at T1h.

Finally, the USDOU did not show significant differences when comparing ADM and ADS treatments and did not differ between intervals within each group. Similar results were observed after the administration of romifidine (80 and 120 µg kg^−1^), where contractions of the cecum, right and left ventral colon, and, to a lesser extent, the duodenum were reduced from 15 and 30 min after administration and remained reduced for up to one hour [9]. Our findings may suggest that ultrasonography is a more sensitive method for assessing intestinal motility in horses, as it allows the detection of gastrointestinal activity through the measurement of contractions per minute.

In the present study, horses in the ADM group showed significant stomach dilation compared to the ADS group over time. However, this dilation was not considered pathological, as it did not result in abdominal discomfort and was not associated with an increase in heart rate (HR) or respiratory rate (fR), making it clinically irrelevant. Nevertheless, it is recommended to monitor horses after opioid administration for stomach distention and to control feeding during the first few hours to avoid complications considering the findings of this study.

In other species, the effects of µ, κ, and δ opioid agonists on gastrointestinal motility and feeding behavior have been well documented [37,38], showing the involvement of central µ opioid receptors in animals with compulsive eating behavior. However, limited data are available on the effects of morphine [7], and no data exist on the effects of methadone on eating and drinking behavior in horses. Regarding morphine, a study evaluated the stomach size of horses that received 0.1 mg/kg^−1^ of morphine, administered three times a day (on Day 2), with 4 h intervals between doses, recording the amounts of hay and water consumed [7]. This study revealed that stomach size significantly increased with the cumulative effect of repeated doses of morphine. Hay consumption (+0.4 kg h^−1^, *p* < 0.001) and water (+1.1 L h^−1^, *p* < 0.001) significantly increased compared to the control days (Days 1 and 3). In the present study, the patients had access to two kilograms of hay 3 h after the procedure. However, water was offered ad libitum, which may have contributed to the stomach distension in the patients of the ADM group. The average (s.d.) of the most caudal EIC where the dorsal aspect of the greater curvature of the stomach was visible was 9.94 (1.25) on Day 1, 12.27 (2.05) on Day 2, and 10.18 (2.07) on Day 3, where the stomach returned to baseline values. In the present study, the stomach showed distension after methadone administration; however, the maximum size observed was 15 (13–15) EIC in the ADM group, compared to 11 (10–13) EIC in the ADS group, 8 h after the administration of the protocols.

The present study did not assess stomach size after 8 h, so the exact time when the stomach returned to its normal size is unknown. No complications were reported up to 48 **h** after the procedure. Our results also align with a recent study [39], where nasogastric intubation and water administration resulted in an increase in stomach size on ultrasonography in both fed and unfed horses (14.4 [12,13,14,15,16] IC spaces). The same study concluded that the increase in stomach size in recently fed horses or those under nasogastric intubation may result in gastric distention not related to pathological conditions or colic and should be interpreted accordingly. Additionally, in adult horses, the cranial and caudal dimensions of the stomach can vary, with the cranial dimension found to be between 6 and 10 IC spaces and the caudal dimension between 10 and 15 IC spaces [40]. Our findings suggest that stomach dilation occurred only after the ingestion of water and hay. Furthermore, the maximum displacement observed was 15 IC spaces, which is consistent with the literature.

The present study has some limitations, including the small number of animals, which prevented the inclusion of additional experimental groups; the lack of a sedation scale, which would have allowed for a greater difference to be observed between the protocols; and the absence of additional scales or cortisol measurements for pain assessment in horses. Post hoc test analysis revealed test powers ranging from 0.1 to 0.8. The lack of statistical differences between protocols for some ultrasound measurements, such as USDOU at T6h and T8h, may be attributed to low power in these comparisons (power values of 0.33 and 0.49, respectively). However, no significant median differences were observed. The similarity in values suggests any difference is unlikely to be clinically relevant. The combination of phenothiazine, opioids, and α2-adrenergic agonists, known as neuroleptanalgesia, enhances the sedative and analgesic effects of each drug, allowing for reduced doses and minimizing side effects [3,4]. This approach increases the safety of the procedures performed in the quadrupedal position in horses. This study suggests that the combination of methadone with detomidine and acepromazine may be beneficial in equines, as the opioid both potentiates and prolongs the analgesia induced by detomidine without causing clinically relevant side effects in heart rate, respiratory rate, or intestinal motility.

## 5. Conclusions

The combination of methadone (0.05 mg kg^−1^) with acepromazine and detomidine (ADM) proved to be a viable neuroleptanalgesia protocol for clinical use in horses. This combination resulted in lower pain scores on the facial expression scale and caused transient intestinal hypomotility. Additionally, this study demonstrated an increase in the greater curvature of the stomach on ultrasound, with no significant adverse effects observed for the horses.

## Figures and Tables

**Figure 1 animals-15-02358-f001:**
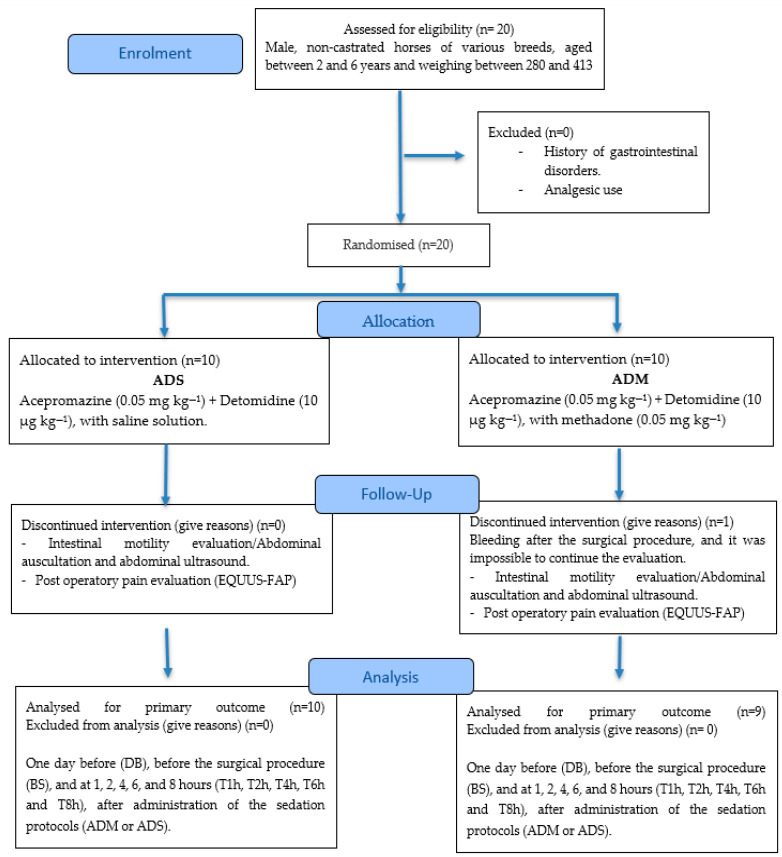
CONSORT 2025 flow diagram.

**Figure 2 animals-15-02358-f002:**
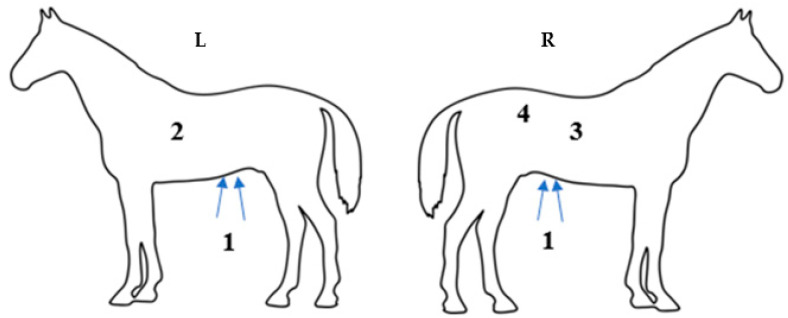
Distribution of ultrasonographic windows in the equine abdomen. (**L**) left side; (**R**) right side. (1) Ventral abdomen—for evaluation of the left and right ventral colon (LVC and RVC); (2) gastric window; (3) duodenal window; (4) paralumbar window—for evaluation of the cecum.

**Table 1 animals-15-02358-t001:** Mean values (±standard deviation) for heart rate (HR), respiratory rate (fR), rectal temperature (RT), and capillary refill time (CRT), in function of time and group. The acronyms (ADM, ADS) represent the groups, and DB, BS, T1h, T2h, T4h, T6h, and T8h represent the different evaluation moments: DB: 1 day before the procedure; BS: before the surgical procedure; T1h, T2h, T4h, T6h, and T8h: 1, 2, 4, 6, and 8 h after administration of 0.05 mg/kg methadone or 0.9% NaCl, respectively. Distinct uppercase letters identify differences between protocols (groups), and distinct lowercase letters indicate differences between the evaluated time points (*p* < 0.05).

Parameter	Group	N	Time (T)						
			DB	BS	T1h	T2h	T4h	T6h	T8h
**HR**	ADM	10	39 (6.9) ^Aa^	31 (4.0) ^Ac^	32 (5.5) ^Ac^	31 (3.8) ^Ac^	33 (5.2) ^Abc^	35 (3.6) ^Aab^	37 (4.5) ^Aa^
	ADS	9	42 (15.0) ^Aa^	40 (12.1) ^Aabc^	34 (6.4) ^Ac^	34 (8.0) ^Ac^	36 (7.4) ^Abc^	42 (8.5) ^Abc^	42 (8.9) ^Aa^
**fR**	ADM	10	24 (9.0) ^Aa^	18 (5.0) ^Ab^	14 (4.2) ^Ac^	13 (2.8) ^Ac^	13 (3.3) ^Ac^	12 (2.7) ^Ac^	13 (4.2) ^Ac^
	ADS	9	21 (4.4) ^Aa^	17 (10.4) ^Ab^	9 (3.2) ^Ac^	8 (2.2) ^Ac^	12 (6.3) ^Abc^	15 (8.6) ^Abc^	15 (7.7) ^Abc^
**RT**	ADM	10	37.8 (0.3) ^Ab^	37.2 (0.5) ^Ac^	36.4 (0.3) ^Ad^	36.4 (0.4) ^Ad^	37.4 (0.7) ^Ac^	38.0 (0.7) ^Aab^	38.2 (0.7) ^Aa^
	ADS	9	37.7 (0.2) ^Ab^	37.4 (0.4) ^Ac^	36.7 (0.8) ^Ad^	36.7 (0.5) ^Ad^	37.2 (0.4) ^Ac^	37.9 (0.6) ^Aab^	38.3 (0.4) ^Aa^
**CRT**	ADM	10	2 (0.0) ^Aa^	2 (0.0) ^Aa^	2 (0.0) ^Aa^	2 (0.3) ^Aa^	2 (0.4) ^Aa^	2 (0.0) ^Aa^	2 (0.0) ^Aa^
	ADS	9	2 (0.0) ^Aa^	2 (0.0) ^Aa^	2 (0.0) ^Aa^	2 (0.0) ^Aa^	2 (0.0) ^Aa^	2 (0.3) ^Aa^	2 (0.0) ^Aa^

**Table 2 animals-15-02358-t002:** Median values (Q1–Q3) of intestinal motility, as obtained by auscultation of the four quadrants: left dorsal quadrant (LDQA), left ventral quadrant (LVQA), right dorsal quadrant (RDQA), and right ventral quadrant (RVQA), as a function of time and groups. Distinct uppercase letters indicate differences between protocols (groups), and distinct lowercase letters indicate differences between the evaluated time points. DB: 1 day before the procedure; BS: before the surgical procedure; T1h, T2h, T4h, T6h, and T8h: 1, 2, 4, 6, and 8 h after the administration of 0.05 mg/kg of methadone or 0.9% NaCl, respectively.

Parameter	Group	N	Time (T)						
			DB	BS	T1h	T2h	T4h	T6h	T8h
**LDQA**	ADM	10	2 (2–2) ^Aab^	2 (2–2) ^Aabc^	1.5 (1–2) ^Ac^	2 (1–2) ^Abc^	2 (2–2.75) ^Aa^	2 (2–2) ^Aabc^	2 (2–2) ^Aabc^
	ADS	9	2 (2–2) ^Aa^	2 (2–2) ^Aa^	1 (0–1) ^Bc^	1 (1–2) ^Ab^	2 (2–2) ^Ba^	2 (1–2) ^Aab^	2 (2–2) ^Aab^
**LVQA**	ADM	10	2 (2–2) ^Ab^	2 (2–2) ^Ab^	1 (1–1.75) ^Ac^	2 (2–2) ^Ab^	3 (2–3) ^Aa^	2 (2–2) ^Ab^	2 (2–2) ^Ab^
	ADS	9	2 (2–2) ^Aa^	2 (2–2) ^Aa^	1 (1–1) ^Ac^	1 (1–2) ^Bbc^	2 (2–2) ^Bab^	2 (2–2) ^Aa^	2 (2–2) ^Aa^
**RDQA**	ADM	10	2 (2–2) ^Aa^	2 (2–2) ^Aa^	1 (1–1) ^Ab^	2 (2–2) ^Aa^	2 (2–2) ^Aa^	2 (2–2) ^Aa^	2 (2–2) ^Aa^
	ADS	9	2 (2–2) ^Aa^	2 (2–2) ^Aa^	1 (0–1) ^Bb^	1 (1–2) ^Aa^	1 (1–2) ^Ba^	2 (2–2) ^Aa^	2 (2–2) ^Aa^
**RVQA**	ADM	10	2 (2–2) ^Aa^	2 (2–2) ^Aa^	1 (1–2) ^Ab^	2 (2–2) ^Aab^	2 (2–2) ^Aa^	2 (2–2) ^Aa^	2 (2–2) ^Aa^
	ADS	9	2 (2–2) ^Aa^	2 (2–2) ^Aa^	1 (1–1) ^Bb^	1 (1–2) ^Aab^	2 (1–2) ^Aa^	2 (1–2) ^Aa^	2 (2–2) ^Aa^

**Table 3 animals-15-02358-t003:** Median values (Q1–Q3) for the number of contractions per minute in the different abdominal windows evaluated by ultrasonography. Right ventral colon ultrasound and left ventral colon, respectively (USRVC), (USLVC), cecum ultrasound (USCEC), and duodenum ultrasound (USDOU) in function of time and groups. Distinct uppercase letters identify differences between protocols (groups), and distinct lowercase letters indicate differences between the evaluated time points (*p* < 0.05). DB; 1 day before the procedure; BS: before the surgical procedure; T1h, T2h, T4h, T6h, and T8h: 1, 2, 4, 6, and 8 h after administration of 0.05 mg/kg methadone or 0.9% NaCl, respectively.

Parameter	Group	N	Time (T)						
			DB	BS	T1h	T2h	T4h	T6h	T8h
**USRVC**	ADM	10	2 (2–2) ^Aa^	2 (2–2) ^Aa^	2 (2–2) ^Aa^	2 (1–2) ^Aa^	2 (2–2) ^Aa^	2 (2–2) ^Aa^	2 (2–2) ^Aa^
	ADS	9	2 (2–3) ^Aa^	2 (1–2) ^Aa^	2 (2–2) ^Aa^	2 (2–3) ^Aa^	2 (2–2) ^Aa^	1 (1–2) ^Ba^	2 (2–2) ^Aa^
**USLVC**	ADM	10	2 (2–2) ^Aa^	2 (2–2) ^Aa^	2 (1–2) ^Aa^	2 (1.25–2) ^Aa^	2 (2–2) ^Aa^	2 (2–2) ^Aa^	2 (2–2) ^Aa^
	ADS	9	2 (2–3) ^Aa^	2 (1–2) ^Aab^	1 (1–2) ^Ab^	2 (1–2) ^Aab^	2 (1–2) ^Aab^	2 (2–2) ^Aab^	2 (2–2) ^Aab^
**USCEC**	ADM	10	2.5 (2–3) ^Aa^	2 (2–2) ^Aab^	1 (1–1.75) ^Ab^	2 (2–2.75) ^Aab^	2.5 (2–3) ^Aab^	2 (2–3) ^Aa^	2 (2–2) ^Aab^
	ADS	9	2 (1–3) ^Aa^	2 (1–2) ^Aa^	1 (1–2) ^Aa^	2 (1–2) ^Aa^	2 (1–2) ^Aa^	1 (1–2) ^Ba^	2 (2–3) ^Aa^
**USDOU**	ADM	10	2.5 (2–3) ^Aa^	3 (2–3) ^Aa^	2 (2–2) ^Aa^	2 (2–2) ^Aa^	2.5 (2–3) ^Aa^	2 (2–3.5) ^Aa^	3 (2.25–3) ^Aa^
	ADS	9	3 (3–3) ^Aa^	3 (2–3) ^Aa^	2 (2–3) ^Aa^	2 (2–3) ^Aa^	2 (2–3) ^Aa^	3 (2–3) ^Aa^	2 (2–3) ^Aa^

**Table 4 animals-15-02358-t004:** Median values (Q1–Q3) for intercostal space (IC) where the stomach was visualized (ST) by ultrasonography. In function of time and groups. Distinct uppercase letters identify differences between protocols (groups), and distinct lowercase letters indicate differences between the evaluated time points (*p* < 0.05). DB: 1 day before the procedure; BS: before the surgical procedure; T1h, T2h, T4h, T6h, and T8h: 1, 2, 4, 6, and 8 h after administration of 0.05 mg/kg methadone or 0.9% NaCL, respectively.

Time (T)									
Parameter	Group	N	DB	BS	T1h	T2h	T4h	T6h	T8h
IC ST	ADM	10	12.5 (12–14) ^Ab^	10 (9.25–11) ^Ac^	10 (10–10.75) ^Ac^	10 (10–10) ^Ac^	13 (11.25–13.75) ^Ab^	13 (12–14) ^Ab^	15 (13–15) ^Aa^
IC ST	ADS	9	10 (10–11) ^Bab^	9 (8–10) ^Ac^	9 (8–11) ^Abc^	9 (8–10) ^Ac^	10 (9–11) ^Babc^	11 (10–13) ^Ba^	11 (10–13) ^Ba^

**Table 5 animals-15-02358-t005:** Median values (Q1–Q3) for the evaluation of pain by facial expression scale EQUUS-FAP from 0 to 18 points, in function of time and groups. Distinct uppercase letters identify differences between protocols (groups), and distinct lowercase letters indicate differences between the evaluated time points (*p* < 0.05). DB: 1 day before the procedure; BS: before the surgical procedure; T1h, T2h, T4h, T6h, and T8h: 1, 2, 4, 6, and 8 h after administration of 0.05 mg/kg methadone or 0.9% NaCL, respectively.

Parameter	Group	N	Time						
			DB	BS	T1h	T2h	T4h	T6h	T8H
**EQUUS-FAP**	ADM	10	0 (0–0) ^Abc^	0 (0–0) ^Ac^	2 (2–3) ^Aa^	1 (0–2) ^Aa^	0 (0–0) ^Ac^	0 (0–0) ^Ac^	0 (0–0) ^Ac^
	ADS	9	0 (0–0) ^Ad^	0 (0–1) ^Acd^	3 (1–4) ^Aa^	3 (3–4) ^Ba^	1 (1–2) ^Bab^	1 (0–2) ^Bbc^	1 (0–2) ^Abcd^

## Data Availability

Additional data supporting this article are available at http://hdl.handle.net/1843/60842, accessed on 4 August 2025.

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
