# Peer review of "Analgesic and Gastrointestinal Effects of Methadone in Horses Undergoing Orchiectomy"

_animals, 2025, doi:10.3390/ani15162358_

Round 1
Reviewer 1 Report (Previous Reviewer 2)
Comments and Suggestions for Authors
Dear Authors,
You paper is much improved from the first time I read it, The changes you made and addition seem to work etter now.
I still have a few comments our questions Here they are
Mat & Methods
Lines 82 - 83: put "client-owned" between "non-castrated" and "horses" and removed it from line 83.
Lines 84 - 86: I was wondering if we could rewrite those 2 lines. They seem repetitive. For exaple: "Horses had no history of gastro intestinal disorders and were considered healthy according to the ......and biochemical evaluations."
Line 189: Rescue analgesia. Once a horse was rescued, did it stay on the study? What was the protocol related to this kind of situation?
Results
In your tables, when representing significant differences, I understand the capital letter vs the lower case, but in regard to the lowercase, I am confused what a, b, c and d refer to. Could we have an explanation. I thought a would be the first time point, b the second etc... but looking at the results that does not make sense.
Discussion
Line 378: the word "gut" seems to be missing at the start of the sentence. I am sure you mean gut motility.
Author Response
Thank you very much for taking the time to review this manuscript.
Please find attached our detailed response to your comments.

Reviewer 2 Report (New Reviewer)
Comments and Suggestions for Authors
accepted
Author Response
Thank you very much for taking the time to review this manuscript. We truly value and appreciate receiving positive feedback.
Reviewer 3 Report (New Reviewer)
Comments and Suggestions for Authors
Comments to the Author
This study reports on the analgesic and gastrointestinal effects of methadone in horses undergoing orchiectomy. Although opioids have been shown to have analgesic effects in horses, the gastrointestinal side effects remain an issue, and this study, which shows that these effects were minimal, provides useful information in this field. This paper is worthy of publication, but some areas require revision. Therefore, we recommend that you resubmit it, taking the following points into consideration.
L72
Please provide a rationale for how the low dose of methadone was determined in this study, 0.05 mg kg-1. The dose of methadone is important in this study.
Table1
Why is the number of excluded cases "Excluded (n=0)" 0, but the number of randomized cases is 19 "Randomised (n=19)"?
L308
Opioids increase HR and fR, but this study states that methadone administration decreased HR and fR, so please add a discussion of this.
L428
Since no studies were found on the use of methadone in horses to assess intestinal motility using ultrasound, it is best to state that the effects were minimal at the doses used in this study.
Author Response
Thank you very much for taking the time to review this manuscript.

This manuscript is a resubmission of an earlier submission. The following is a list of the peer review reports and author responses from that submission.
Round 1
Reviewer 1 Report
Comments and Suggestions for Authors
Thank you for submitting your manuscript evaluating two sedative protocols for standing procedures in horses.
While the study addresses a clinically important topic, several aspects need substantial improvement. The main limitation lies in the timing of outcome assessment, which starts post-procedure rather than post-drug administration, complicating the interpretation of drug effects. The manuscript would benefit from clearer methodological descriptions, including justification for sample size, sourcing of animals, and statistical approaches that account for repeated measures. Additionally, the formatting, terminology, and organization suggest it was prepared for a different journal and not fully adapted to Animals' guidelines. Simplifying the results, clarifying tables and figures, and streamlining the discussion to focus on findings relevant to your hypothesis would greatly improve the clarity and scientific value of your work. Please, see my comments attached.

The English language is not used correctly. While the manuscript is readable, there are several typos ed editing issues to be addressed. Morever, certain parts are cumbersome.
Author Response
Dear Reviewer, please find attached a document with the responses to your comments.
Thank you very much for your valuable contribution to this manuscript.

Reviewer 2 Report
Comments and Suggestions for Authors
Dear authors,
Thank you for working on this subject. I found your paper interesting.
See my comments in the attached document.

Author Response
Dear Reviewer, please find attached a document with the responses to your comments.
Thank you very much for your valuable contribution to this manuscript."
